# Effect of Cobalt and Chromium Ions on the Chlorhexidine Digluconate as Seen by Intermolecular Diffusion

**DOI:** 10.3390/ijms222413266

**Published:** 2021-12-09

**Authors:** Sónia I. G. Fangaia, Pedro M. G. Nicolau, Fernando A. D. R. A. Guerra, M. Melia Rodrigo, Gianluca Utzeri, Ana M. T. D. P. V. Cabral, Artur J. M. Valente, Miguel A. Esteso, Ana C. F. Ribeiro

**Affiliations:** 1Institute of Implantology and Prosthodontics, Faculty of Medicine, University of Coimbra, 3000-075 Coimbra, Portugal; sfangaia@fmed.uc.pt (S.I.G.F.); pgnicolau@mail.telepac.com (P.M.G.N.); fguerra@ci.uc.pt (F.A.D.R.A.G.); 2U.D. Química Física, Universidad de Alcalá, 28805 Alcalá de Henares, Spain; mmelia.rodrigo@uah.es (M.M.R.); mangel.esteso@ucavila.es (M.A.E.); 3Department of Chemistry, CQC, University of Coimbra, 3004-535 Coimbra, Portugal; gianlucautz@gmail.com (G.U.); anacfrib@ci.uc.pt (A.C.F.R.); 4Faculty of Pharmacy, CQC, University of Coimbra, 3000-295 Coimbra, Portugal; acabral@ff.uc.pt; 5Universidad Católica Santa Teresa de Jesús de Ávila, 05005 Ávila, Spain

**Keywords:** artificial saliva, cobalt chloride, chromium chloride, diffusion coefficients, chlorhexidine digluconate

## Abstract

Metal ions such as cobalt (II) and chromium (III) might be present in the oral cavity, as a consequence of the corrosion of Co-Cr dental alloys. The diffusion of such metal ions into the organism, carried by saliva, can cause health problems as a consequence of their toxicity, enhanced by a cumulative effect in the body. The effect of the chlorhexidine digluconate, which is commonly used in mouthwash formulations, on the transport of these salts is evaluated in this paper by using the Taylor dispersion technique, which will allow an assessment of how the presence of chlorhexidine digluconate (either in aqueous solution or in a commercial formulation) may affect the diffusion of metal ions. The ternary mutual diffusion coefficients of metal ions (Co and Cr) in the presence of chlorhexidine digluconate, in an artificial saliva media, were measured. Significant coupled diffusion of CoCl_2_ (and CrCl_3_) and chlorhexidine digluconate is observed by analysis of the non-zero values of the cross-diffusion coefficients, *D*_12_ and *D*_21_. The observed interactions between metal ions and chlorhexidine digluconate suggest that the latter might be considered as an advantageous therapeutic agent, once they contribute to the reduction of the concentration of those ions inside the mouth.

## 1. Introduction

The use of metal alloys in the manufacture of dental prostheses has been a common practice for decades. Several dental alloys have been used in the fabrication of these devices, such as titanium alloys, cobalt-chromium alloys, chromium-nickel alloys, etc. Although these alloys are biocompatible and relatively resistant to corrosion, when exposed to certain media (e.g., acid environments [1,2]), and also as a consequence of wear [3] resulting from chewing, the release of metal ions in the oral cavity is a matter of concern. If these metal ions are constantly diffusing into body fluids, their concentration in the body will increase and, after a certain latency period, it can reach a toxic threshold value that can cause significant harmful effects on the body (by combining with biomolecules such as enzymes and proteins), leading to health problems [4,5].

The chromium released during the degradation of the Co-Cr alloys is essentially Cr(III), but it can be oxidized to Cr(VI) at the cellular level. Cr(VI) is mutagenic and carcinogenic, with its potential biological effects being controversial [6]. In any case, metals such as chromium and cobalt, which may have potential or demonstrated oncogenic effects in humans, must be subject to strict regulations for the protection of humans [7,8].

The release of metal ions from orthodontic devices in the presence of mouthwashes is a matter of concern given the potential toxicity raise and function loss of devices [9,10,11]. 

Among those devices, prosthetic restorations are subject to fretting corrosion and wear [12,13,14], and the released wear debris may produce ionic species which have a potential toxicity depending on their concentration [15]. Consequently, it is of utmost importance to analyze the behavior of those ions in the oral cavity. There are several studies that identify released elements or determine ion concentrations [2,15]. However, to the best of our knowledge, the use of mass transport by diffusion to assess the interaction between metal ions and mouthwashes is an innovative approach.

In the present manuscript, the interaction between metal ions, in particular divalent cobalt and trivalent chromium ions, and a pharmacological molecule: chlorhexidine digluconate (C_34_H_54_Cl_2_N_10_O_14_, Figure 1) (CHDG), which is present in many mouthwashes, is evaluated by measuring intermolecular diffusion coefficients using the Taylor dispersion method [16,17,18]. We intend to conclude about the potential of CHDG to act as a carrier of those ions, facilitating their removal from the oral cavity and, thus, reducing the potential toxicity.

It is known that a considerable decrease in salivary pH occurs after drinking acidic beverages whose pH is around 2 [19,20], and several studies are related to the behavior of metal alloys under physiological conditions using a pH = 2.3 [1,21].

In the first part of this study, tracer diffusion coefficients, *D*, or apparent diffusion coefficients for aqueous solutions of cobalt (II) chloride and chromium (III) chloride at 0.001 mol dm^−3^, in artificial saliva at pH: 8.3, 7.0, and 2.3, and in artificial saliva with sodium fluoride at pH = 7.0, are reported. This study is complemented by the evaluation of the diffusion behavior of the chlorhexidine digluconate (as a pure compound or contained in commercial formulation) in aqueous solution in the absence or presence of metal ions’ salts. UV-visible spectroscopy was also used to assess the effect of chlorhexidine digluconate on Co(II) and Cr(III).

## 2. Results and Discussion

Table 1 shows the tracer diffusion coefficients, *D*^0^_1_, for aqueous solutions of cobalt (II) chloride and chromium (III) chloride in water and in CHDG-containing solutions. It can be seen that the *D*^0^_1_ value for CoCl_2_ matches with that previously reported in [22] and measured by using the open-ended capillary cell. Although the tracer diffusion coefficients are similar for both salts, and for the mixture as well, at pH 6.4, the predominant cationic species are Co^2+^ [23] and Cr_3_(OH)_4_^5+^ [24]. It can also be seen that, in general, the tracer diffusion coefficients decrease by decreasing the solution pH. In fact, the presence of GHDC either as pure or in a commercial formulation leads to a slight increase in the acidity of the media. 

Table 2 shows the tracer diffusion coefficients of the same salts and mixture of salts in artificial saliva (AS). In order to evaluate the effect of ionic strength and pH on the tracer diffusion coefficients, we have used sodium fluoride and lactic acid, to control both parameters, respectively. The former has been chosen as it is an important component of several oral mouth rinses used for preventing dental caries [25,26], whilst the latter has been used to simulate the variation of pH occurring in the oral cavity [27,28].

Table 3 shows the average experimental diffusion coefficients of solutions of chlorhexidine digluconate (*D*_CHDG_). These values are compared with those obtained by using a CHDG-based commercial formulation (*D*_CHDG-cf_). It should be noted that information on apparent diffusion coefficients of CHDG in a commercial mixture is of significant importance for understanding the behavior of this compound in practical dental applications. The reproducibility of these diffusion coefficients is similar to that observed for other systems, i.e., within ±2% [29,30].

Tracer diffusion coefficients of CoCl_2_ and CrCl_3_ and the mixture of them increase significantly in artificial saliva and in artificial saliva with NaF pH = 7.0, when compared with those obtained in water. The deviations between the tracer diffusion coefficient values (*D*^0^_1_) in these media and the limiting diffusion coefficients of these salts in water (*D*^0^) [31,32], at the same temperature, are positive (Δ*D*^0^/*D*^0^ = 44%, 62%, and 95%, for CoCl_2_ and CrCl_3_, and the mixture of them, respectively). The increase in these *D*^0^ values when compared with the *D*^0^ value in water indicates the presence of salting-out effects. These ions, as chlorides, will suffer less frictional resistance to motion through the fluid and, consequently, their diffusion coefficients in these media become higher, and can flow faster inside living tissues, which can cause severe disturbances associated with these heavy metal ions.

However, in acidic media (artificial saliva pH = 2.3 with and without NaF), tracer diffusion coefficient values are much smaller when compared with *D*^0^ values in water. This fact may be explained on the basis of an electrostatic mechanism. Considering that in acid solutions, H_3_O^+^ is one of the predominant species, due to its large mobility, a strong electric field will be generated by a concentration gradient in H^+^. Slowing down these H^+^ ions drives large counter-current fluxes of Co^2+^ and Cr^3+^ in aqueous solutions, and consequently, their values of *D* < 0 (salting-in). For Cr(III) solutions, an alteration in the structure of the solution due to modification of predominant species, as a function of pH, is also expected.

Table 3 shows the average experimental diffusion coefficients of chlorhexidine digluconate in water and in a commercial formulation (cf). It is observed that the diffusion coefficients of CHDG at cf are higher than in water. In the case of salts, it has been found that pH is slightly lower than water and that has an effect in the tracer diffusion coefficients. Another possible explanation comes from the occurrence of higher ionic strength in the cf, thus inducing less electrostatic interactions between the diffusing species [33]. It can also be noticed that, in both cases, diffusion coefficients decrease by increasing the concentration; however, the effect of concentration in the diffusion of CHDG in the cf is more significant, when compared with water.

These results show that, although the interpretation of diffusion coefficients is not straightforward, they can provide relevant information on the effect of different conditions on the mobility of relevant ions, which can be of utmost importance for practical purposes. Keeping that in mind, ternary mutual diffusion coefficients for the systems {(CoCl_2_) (1) + CHDG (2)} and {(CrCl_3_) (1) + CHDG (2)} in water and in a commercial formulation (cf), were measured at tracer concentrations, and data are reported in Table 4 and Table 5, respectively.

Tracer diffusion coefficient values for chlorhexidine digluconate (at mole fraction *X*_2_ = 0) are reported in Table 4 and Table 5. It should be noted that the main diffusion coefficients *D*_11_ and *D*_22_ were generally reproducible within ±0.02 × 10^−9^ m^2^ s^−1^, whilst the cross-coefficients were reproducible within ±0.05 × 10^−9^ m^2^ s^−1^.

The *D*_11_ values are considerably larger than the *D*_22_ values, and, in the case of CrCl_3_, they are higher than the binary diffusion coefficient of this salt measured previously by the same technique [31,32].

It can also be seen that coupled diffusion is significant. Indeed, *D*_21_/*D*_11_ ratios indicate that a mole of diffusing salt can counter-transport up to 0.6 mol of chlorhexidine digluconate, whereas the values of the ratio *D*_12_/*D*_22_ show that a mole of diffusing chlorhexidine can counter-transport up to 1.2 mol of salt (CrCl_3_). 

From the significant negative cross-diffusion coefficients for the system CrCl_3_ and chlorhexidine, indicating counter-current coupled flows of this salt and chlorhexidine digluconate, we can infer that there is evidence of strong interactions between these two components. This mechanism also accounts for the large negative values of cross-diffusion as well as the increased diffusivity of the CrCl_3_ and CoCl_2_ components in aqueous chlorhexidine digluconate (Table 4 and Table 5). The presence of complex ions between Co(II) and Cr(III) ions and chlorhexidine digluconate may explain the obtaining of negative *D*_12_ values. That is, in the region of the solution where the concentration of chlorhexidine digluconate is highest, there will be a more pronounced decrease of a large amount of the free Co^2+^ and Cr^3+^ ions resulting from the formation of those complex ions, hence the appearance of a gradient of these ions (Co(II) or Cr(III)) with a signal opposite to the gradient of cobalt chloride (or chromium chloride).

Support for this effect of chlorhexidine digluconate on chromium ions was further assessed by UV-visible spectroscopy. Figure 2a shows the spectra of aqueous solutions of CrCl_3_·6H_2_O. The spectra show characteristics of maximum absorbances at 417 and 520 nm [34]. These spectra are similar to those obtained for Cr(III) in the presence of CHDG (1 mM). However, an increase in the absorbance was found at both wavelengths: 417 and 590 nm (Figure 2b), accompanied by a maximum 0.2 and 0.05 red-shift displacement. No alteration in the spectra of Co(II) in the absence and presence of CHDG was observed (not shown). This can be justified by changes in the conjugated system of ligand molecules. The Cr(III) water exchange rate, from the first hydration shell, is very low [23], which confers some inert features for ligand substitution [35]; consequently, it can be hypothesized that a strong interaction with some component of CHDG, acting as a ligand, takes place. Concomitantly, Cr(III) has high charge density when compared with Co(II) (the ionic radii of Co(II) and Cr(III) are 0.74 and 0.61 nm, respectively [36]). The digluconate shows a high affinity towards metal ions (for example, calcium and aluminum) and, consequently, it could be the referred ligand [37,38]. 

## 3. Materials and Methods

### 3.1. Materials

Cobalt(II) chloride hexahydrate (Sigma-Aldrich, pro-analysis, purity (mass fraction) >0.99, Lyon, France), chromium(III) chloride hexahydrate (Riedel-de-Haen, pro-analysis >97%, Seelze, Germany), and chlorhexidine digluconate solution (20% in water, Sigma-Aldrich, Lyon, France) were used as received without further purification (Table 6).

The solutions needed for diffusion measurements were prepared in calibrated volumetric glass flasks, using as solvents: ultrapure water (Millipore, Germany, Milli-Q Advantage A10, specific resistance = 1.82 × 10^5^ Ω m, at 298.15 K), artificial saliva, prepared according the composition indicated in Table 6, and a commercially available Extra Eludril^®^ from Pierre Fabre Oral Care, with 0.2% chlorhexidine digluconate.

The weighing was performed using a Radwag AS 220C2 balance with readability of 10^−5^ g in the lower range.

The pH measurements of solutions were carried out with a Radiometer pH meter PHM 240 with an Ingold U457-K7pH conjugated electrode. pH was measured in fresh solutions, and the electrode was calibrated immediately before each experimental set of solutions using IUPAC-recommended pH 4, 7, and 10 buffers. From pH meter calibration, a zero-pH of 6.897 ± 0.030 and sensitivity higher than 98.7% were obtained. To perform these measurements at pH 2.3 and 7.0, the intended values of the pH were adjusted by the addition of lactic acid. All solutions were freshly prepared at 298.15 K and degassed by sonication for about 60 min before each experiment.

### 3.2. Taylor Method

#### 3.2.1. Tracer Diffusion Coefficients

We consider the system containing cobalt chloride (or chromium chloride) in artificial saliva as a pseudo-binary system. That is, a system with two components, Co(II) or Cr(III), and Cl^−^ ions, assuming the artificial saliva (with and without lactic acid and sodium fluoride components) as a mixed solvent. 

The diffusion coefficient, *D*, in these pseudo-binary systems may be defined in terms of the concentration gradient by the phenomenological relationship of Fick’s first law (Equation (1)):(1)J=−D∇C
where *J* and ∇*C* are the molar flux and the gradient in the concentration of solute, respectively.

In addition, the ionic strength of the artificial saliva is significantly higher than the salt concentration under study (CoCl_2_ 0.001 mol dm^−3^ or CrCl_3_ 0.001 mol dm^−3^) (that is, approximately 0.075 mol dm^−3^), ensuring the occurrence of tracer diffusion, and the composition of saliva in the injection and carrier solutions are equal. This is also confirmed by the detector signal resembling a single normal distribution with variance r^2^t_R_/24*D*_T_, and not two overlapping normal distributions [17]. We may consider the system as pseudo-binary and consequently take the measured parameters as the tracer diffusion coefficients of the CoCl_2_ (or CrCl_3_) in the artificial saliva. 

The Taylor dispersion method is based on the dispersion of small amounts of solution injected into laminar carrier streams of water or solution of different composition, flowing through a long capillary tube (Figure 3). Since the detailed description of this method can be found in the literature (e.g., [17,18,41,42,43]), only a few more relevant points are highlighted here.

The length of the dispersion tube (PTFE) used in the present study was measured directly by stretching the tube in a reservoir and using two high-quality theodolytes and appropriate mirrors to accurately focus on the tube ends. This technique afforded a tube length of 3.2799 (±0.0001) × 10^4^ mm, in agreement with less-precise control measurements using a good-quality measuring tape. The radius of the tube, 0.5570 (±0.00003) mm, was calculated from the tube volume obtained by accurately weighing (resolution 0.1 mg) the tube when empty and when filled with distilled water of known density. The tube was mounted on a rigid cylindrical support in side-by-side coils of 200 mm radius.

At the start of each run, a 6-port poly(tetrafluoroethylene) injection valve (Rheodyne, model 5020) was used to introduce 0.063 mL of solution into the laminar carrier stream of slightly different composition. A flow rate of 0.17 mL min^−1^ was maintained by a metering pump (Gilson model Minipuls 3) to give retention times of about 1.1 × 10^4^ s. The dispersion tube and the injection valve were kept at 298.15 K (±0.01 K) in an air thermostat.

Dispersion of the injected samples was monitored using a differential refractometer (Waters model 2410) at the outlet of the dispersion tube. Detector voltages, *V*(*t*), were measured at precisely 5 s intervals with a digital voltmeter (Agilent 34401 A) with an IEEE interface. Binary diffusion coefficients were evaluated by fitting the dispersion equation:*V*(*t*) = *V*_0_ + *V*_1_*t* + *V*_max_ (*t*_R_/*t*)^1/2^ exp[−12*D*(*t* − *t*_R_)^2^/*r*^2^*t*](2)
to the detector voltages. The additional fitting parameters are the mean sample retention time, *t*_R_, peak height, *V*_max_, baseline voltage, *V*_0_, and baseline slope, *V*_1_.

#### 3.2.2. Diffusion of Chlorhexidine Digluconate

The diffusion behavior of aqueous solutions of chlorhexidine digluconate at different concentrations without added salts was analyzed. The dispersion profiles were prepared by injecting water into carrier streams of chlorhexidine digluconate 0.2%, and by injecting chlorhexidine digluconate of composition 0.2% and 0.04% (*m*/*v*) into water. Although chlorhexidine digluconate 0.2% was accompanied by other substances, the resulting aqueous systems were assumed as pseudo-binaries. The respective diffusion coefficients were measured for the same procedure described in the preceding Section 3.1, and evaluated by fitting the dispersion equation (Equation (2)). 

The effect of chlorhexidine digluconate on the diffusion of cobalt and chromium ions was investigated by using Taylor dispersion to measure the ternary mutual diffusion coefficients (*D*_ik_) of aqueous CoCl_2_ (or CrCl_3_)(*C*_1_) + chlorhexidine digluconate (*C*_2_) solutions and using coupled Fick equations (Equations (3) and (4)) [44,45]:*J*_1_ = −*D*_11_∇*C*_1_ − *D*_12_∇*C*_2_(3)
*J*_2_ = −*D*_21_∇*C*_1_ − *D*_22_∇*C*_2_(4)
where *J*_1_ and *J*_2_ are the molar fluxes of CoCl_2_ (or CrCl_3_) (component 1) and chlorhexidine digluconate (component 2) driven by the concentration gradients, ∇*C*_1_ and ∇*C*_2_, of each solute 1 and 2, respectively. *D*_11_ and *D*_22_ are the main, whilst *D*_12_ and *D*_21_ are the cross-diffusion coefficients. For more details see, for example, [29]. 

In the present work, pseudo-ternary dispersion profiles were prepared by injecting CoCl_2_ (or CrCl_3_) (component 1) + chlorhexidine digluconate (component 2) solution samples of composition ${\bar{C}}_{1}+\Delta C_{1}$, ${\bar{C}}_{2}$ into carrier streams of composition ${\bar{C}}_{2}$, and by injecting CoCl_2_ (or CrCl_3_) (component 1) + chlorhexidine digluconate (component 2) solution samples of composition ${\bar{C}}_{1}$, ${\bar{C}}_{2}+\Delta C_{2}$ into carrier streams of composition ${\bar{C}}_{1}$ Coupled diffusion produces ternary dispersion profiles (Equations (5) and (6)):(5)$$V=\bar{V}+V_{1}+V_{\max}\sqrt{\frac{t_{R}}{t}}\frac{\left(a+b\alpha_{1}\right)\sqrt{D_{1}}e^{-12D_{1}{\left(t-t_{R}\right)}^{2}/r^{2}t}+\left(1-a-b\alpha_{1}\right)\sqrt{D_{2}}e^{-12D_{2}{\left(t-t_{R}\right)}^{2}/r^{2}t}}{\left(a+b\alpha_{1}\right)\sqrt{D_{1}}+\left(1-a-b\alpha_{1}\right)\sqrt{D_{2}}}$$
(6)$$\alpha_{1}=\frac{R_{1}\Delta C_{1}}{R_{1}\Delta C_{1}+R_{2}\Delta C_{2}}$$
where *D*_1_ and *D*_2_ are the eigenvalues of the matrix of ternary *D*_ik_ coefficients (Equations (7) and (8)) and *α*_1_ is the fraction of the initial refractive index difference due to CoCl_2_ (or CrCl_3_). *R*_1_ and *R*_2_ are the detector sensitivities for CoCl_2_ (or CrCl_3_) (1) and chlorhexidine digluconate (2): *R*_1_ = ∂*V*/∂*C*_1_ and *R*_2_ = ∂*V*/∂*C*_2_.
(7)$$D_{1}=\left\{D_{11}+D_{22}+\left(D_{11}-D_{22}\right)\sqrt{1+\left[4D_{12}D_{21}/{\left(D_{11}-D_{22}\right)}^{2}\right]}\right\}/2$$
(8)$$D_{2}=\left\{D_{11}+D_{22}-\left(D_{11}-D_{22}\right)\sqrt{1+\left[4D_{12}D_{21}/{\left(D_{11}-D_{22}\right)}^{2}\right]}\right\}/2$$

Ternary mutual diffusion coefficients were calculated from *D*_1_, *D*_2_, *a*, and *b* fitting parameters and the relative detector sensitivity, *R*_2_/*R*_1_, using:(9)$$D_{11}=D_{1}+\frac{a\left(1-a-b\right)}{b}\left(D_{1}-D_{2}\right)$$
(10)$$D_{12}=\frac{R_{2}}{R_{1}}\frac{a\left(1-a\right)}{b}\left(D_{1}-D_{2}\right)$$
(11)$$D_{21}=\frac{R_{1}}{R_{2}}\frac{\left(a+b\right)\left(1-a-b\right)}{b}\left(D_{2}-D_{1}\right)$$
(12)$$D_{22}=D_{2}+\frac{a\left(1-a-b\right)}{b}\left(D_{2}-D_{1}\right)$$
*a* and *b* parameters in Equations (9)–(12) are described by:(13)$$a=\frac{D_{11}-D_{1}-(R_{1}/R_{2})D_{12}}{D_{2}-D_{1}}$$
(14)$$b=\frac{D_{22}-D_{11}+(R_{1}/R_{2})D_{12}-(R_{2}/R_{1})D_{21}}{D_{2}-D_{1}}$$

### 3.3. UV-Vis Spectroscopy Measurements

Electronic absorption spectroscopy of solutions of Cr(III) and Co(II), in the concentration range 10–50 mM, in water and in a mixture containing chlorohexidine digluconate, was carried out by using a Shimadzu UV-2600i UV-Vis spectrophotometer.

## 4. Conclusions

Binary and ternary diffusion coefficients of cobalt chloride and chromium chloride, and chlorohexidine digluconate, alone and in mix solutions, respectively, were measured by the Taylor dispersion technique. In artificial saliva at pH = 2.3, containing lactic acid and sodium fluoride, a decrease of the diffusion coefficients of these salts from approximately 10% to 40% was observed when compared with those obtained in water. In the presence of saliva, a salting-in effect affecting the metal ion salts was observed. 

However, in artificial saliva at pH = 7 and 8.0, the diffusion coefficients of these salts increased significantly (at most 60%), which indicates the presence of salting-out effects. These ions will suffer less frictional resistance to motion through the fluid and, consequently, their diffusion coefficients in these media become higher and can flow faster inside living tissues, causing severe disturbances associated with these heavy metal ions. 

Interactions between the metal salts and CHDG were also observed, essentially for Cr(III), by UV-Vis spectroscopy.

The experimental results suggested that interactions between metal ions and CHDG might be justified by the occurrence of metal–digluconate interactions and are stronger for Cr(III), probably due to its high charge density.

We can conclude that the chlorhexidine digluconate may be used as a controlled heavy metal chromium and cobalt capture system, and therefore contribute to reduce the toxicity levels in the oral cavity. 

## Figures and Tables

**Figure 1 ijms-22-13266-f001:**
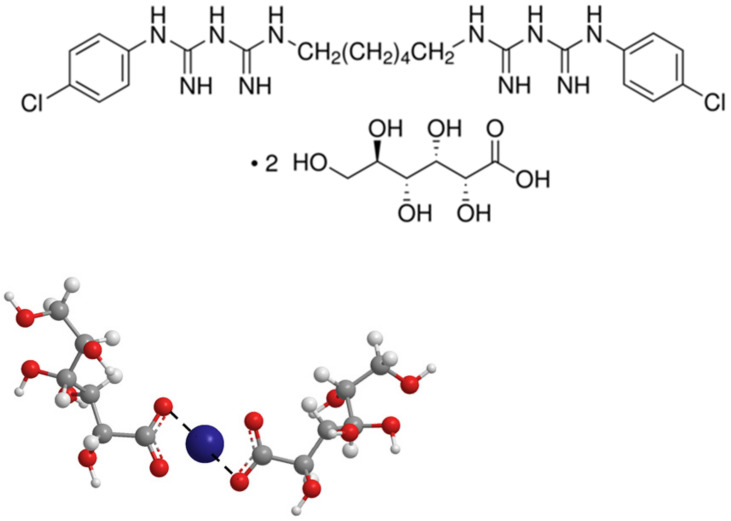
Chlorhexidine digluconate structure (**top**) and schematic representation of complexation between a divalent metal ion (e.g., Co^2+^, blue sphere) and digluconate (**bottom**).

**Figure 2 ijms-22-13266-f002:**
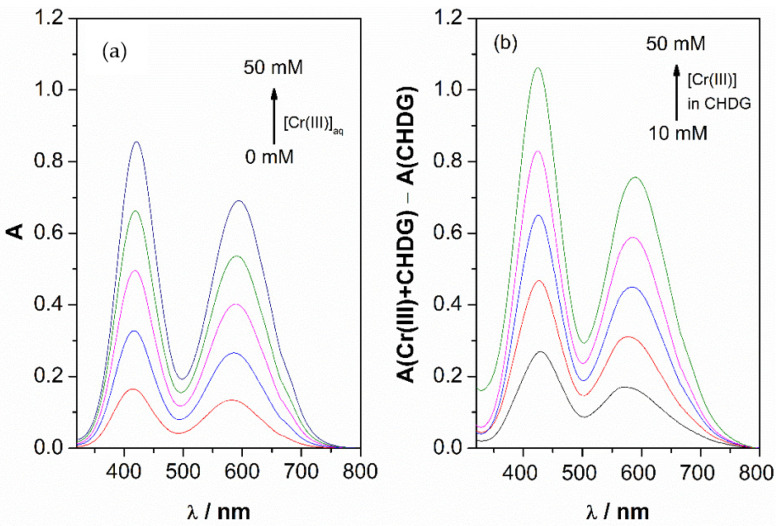
UV-vis spectra of aqueous solution of CrCl_3_ (**a**) and solutions of CrCl_3_ in chlorhexidine digluconate (**b**) (0.004 mol dm^−3^).

**Figure 3 ijms-22-13266-f003:**
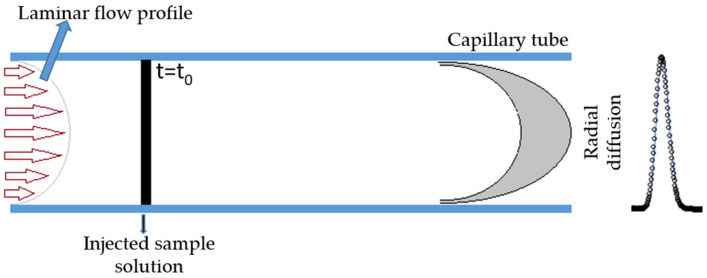
Schematic representation of the dispersion technique.

**Table 1 ijms-22-13266-t001:** Tracer diffusion coefficients, *D*^0^_1_, of CoCl_2_ (0.001 mol dm^−3^), CrCl_3_ (0.001 mol dm^−3^), and a mixture of CoCl_2_ and CrCl_3_ in different aqueous media, and the respective standard deviations of the mean ^(a)^, *S**_D_*, at *T* = 298.15 K and *P* = 101.3 kPa.

		*D*^0^_1_ ± *S_D_*/(10^−9^ m^2^ s^−1^)		
	pH	CoCl_2_	CrCl_3_	CoCl_2_/CrCl_3_
water	6.4	1.295 ± 0.010	1.272 ± 0.014	1.189 ± 0.015
0.2% CHDG	5.0	0.666 ± 0.015)	0.656 ± 0.015	-
CHDG commercial formulation	5.7	0.709 ± 0.020)	0.718 ± 0.018	0.915 ± 0.022

^(a)^ Averaged result for n = 3 experiments. Standard uncertainties, *u,* are *u*_r_(*C*) = 0.03, *u*(*T*) = 0.01 K, and *u*(*P*) = 2.03 kPa.

**Table 2 ijms-22-13266-t002:** Tracer diffusion coefficients, *D*^0^_1_*,* of CoCl_2_, CrCl_3_, and a mixture of CoCl_2_ and CrCl_3_ in different artificial saliva (AS) media, at *T* = 298.15 K and *P* = 101.3 kPa.

		*D*^0^_1_ ± *S_D_*/(10^−9^ m^2^ s^−1^) ^(a)^		
	pH	CoCl_2_	CrCl_3_	CoCl_2_/CrCl_3_
AS	8.3	1.823 ± 0.024	1.789 ± 0.040	1.890 ± 0.045
AS	7.0	1.860 ± 0.010	1.808 ± 0.020	1.909 ± 0.014
AS + lactic acid	2.3	0.921 ± 0.030	0.908 ± 0.012	0.917 ± 0.010
AS + NaF	7.0	1.701 ± 0.029	2.065 ± 0.019	2.315 ± 0.020
AS + lactic acid + NaF	2.3	0.899 ± 0.031	0.782 ± 0.030	0.826 ± 0.028

^(a)^ Averaged result for n = 3 experiments. Standard uncertainties, *u,* are *u*_r_(*C*) = 0.03, *u*(*T*) = 0.01 K, and *u*(*P*) = 2.03 kPa.

**Table 3 ijms-22-13266-t003:** Diffusion coefficients for aqueous chlorhexidine digluconate solutions, *D*_CHDG_, at different concentrations, *C*, at temperature *T* = 298.15 K and pressure *P* = 101.3 kPa.

*C/*(mol dm^−3^)	*D*_CHDG_/(10^−9^ m^2^ s^−1^)	*D*_CHDG,cf_/(10^−9^ m^2^ s^−1^)
0.000 *	0.635 ± 0.010	0.762 ± 0.016
0.001	0.617 ± 0.009	0.740 ± 0.012
0.004	0.602 ± 0.008	0.677 ± 0.013

* There is no CHDG in the flux and, consequently, the tracer diffusion coefficient is measured. The standard uncertainties are *u*_r_(*C*) = 0.03, *u*(*T*) = 0.01 K, *u*(*P*) = 2.03 kPa, and *u*(*D*) = 0.01 × 10^−9^ m^2^ s^−1^.

**Table 4 ijms-22-13266-t004:** Tracer ternary diffusion coefficients (*D*_11_, *D*_12_, *D*_21_, *D*_22_) of aqueous CoCl_2_ (or CrCl_3_) (*C*_1_ = 1 × 10^−3^ mol dm^−3^) + CHDG (*C*_2_ = 0) solutions and at *T* = 298.15 K and *P* = 101.3 kPa.

	*D*_11_ ± *S_D_*	*D*_12_ ± *S_D_*	*D*_21_ ± *S_D_*	*D*_22_ ± *S_D_*	*D*_12_/*D*_22_
CoCl_2_	1.325 ± 0.018	−0.105 ± 0.030	−0.020 ± 0.020	0.780 ± 0.010	−0.135
CrCl_3_	1.310 ± 0.020	−0.205 ± 0.030	−0.050 ± 0.010	0.736 ± 0.010	−0.279

Diffusion coefficients and standard deviation, *S_D_*, in units of 10^−9^ m^2^ s^−1^. The standard uncertainties are *u*_r_(*C*) = 0.03, *u*(*D*) = 0.01 × 10^−9^ m^2^ s^−1^, *u*(*T*) = 0.01 K, and *u*(*P*) = 2.03 kPa.

**Table 5 ijms-22-13266-t005:** Tracer ternary diffusion coefficients (*D*_11_, *D*_12_, *D*_21_, *D*_22_) of aqueous CoCl_2_ (or CrCl_3_) (*C*_1_ = 1 × 10^−3^ mol dm^−3^) + (CHDG,*cf*) (*C*_2_ = 0) solutions and at *T* = 298.15 K and *P* = 101.3 kPa.

	*D*_11_ ± *S_D_*	*D*_12_ ± *S_D_*	*D*_21_ ± *S_D_*	*D*_22_ ± *S_D_*	*D*_12_/*D*_22_	*D*_21_/*D*_11_
CoCl_2_	1.193 ± 0.018	−0.150 ± 0.030	−0.003 ± 0.020	0.809 ± 0.010	−0.060	−0.003
CrCl_3_	1.309 ± 0.020	−1.080 ± 0.030	−0.002 ± 0.010	0.819 ± 0.010	−1.201	−0.001

Diffusion coefficients and standard deviation, *S_D_*, in units of 10^−9^ m^2^ s^−1^. The standard uncertainties are *u*_r_(*C*) = 0.03, *u*(*D*) = 0.01 × 10^−9^ m^2^ s^−1^, *u*(*T*) = 0.01 K, and *u*(*P*) = 2.03 kPa.

**Table 6 ijms-22-13266-t006:** Sample description.

Chemical	Source	CAS Number	Mass Fraction Purity
CoCl_2_·6H_2_O	Sigma-Aldrich	7791-13-1	>0.98 ^(a)^
CrCl_3_·6H_2_O	Riedel-de-Haen, Seelze	10060-12-5	>0.98 ^(a)^
NaF	Sigma-Aldrich		>0.99 ^(a)^
Lactic acid	Sigma-Aldrich		>0.85 wt.% ^(a)^
Artificial saliva ^(b)^			
Chlorhexidine digluconate ^(c)^	Sigma-Aldrich	18472-51-0	20% in water
H_2_O	Millipore-Q water (1.82 × 10^5^ Ω m at 298.15 K)	7732-18-5	

^(^^a)^ As stated by the supplier. ^(^^b)^ Artificial saliva was prepared according to the following composition [39,40]: potassium chloride (KCl, 20 mmol/L), sodium hydrogenocarbonate (NaHCO_3_, 17.9 mmol/L), sodium phosphate (NaH_2_PO_4_·H_2_O, 3.6 mmol/L), potassium thiocyanate (KSCN, 5.1 mmol/L), and lactic acid (0.10 mmol/L). ^(^^c)^ In this work, a commercial formulation containing 0.2% chlorhexidine digluconate was also used.

## Data Availability

Data are contained within the article.

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
