# Peer review of "Effect of Cobalt and Chromium Ions on the Chlorhexidine Digluconate as Seen by Intermolecular Diffusion"

_ijms, 2021, doi:10.3390/ijms222413266_

Round 1

Reviewer 1 Report

This is a very interesting manuscript dealing withe the determination of the diffusion coefficients of metal ions (Co and Cr) in presence of antiseptic agent chlorhexidine digluconate (CHDG) in water and in an artificial saliva media. Results demonstrate the capacity of CHDG to reduce the possible toxic effects of these ions present in some metal alloy dental prostheses. The manuscript could be accepted after checking some minor points as indicated in the attached pdf file.

Author Response

We really thank the Reviewer for his/her careful reading of the ms. All suggestions were taken into account and the ms was improved accordingly.

Reviewer 2 Report

The article is interesting and has promising results for the targeted application, but there is still place for better.

I recommend to the authors to consider the following remarks:

  1. I find the introduction a little bit unclear. Actually, from line 49 to line 73, it’s not clearly highlighted the aim of the study. Try to reorganize it. Also, it is very important to mention in the introduction the novelty of the study, because there are also other studies in the literature focused on the effect of mouthwashes on metallic ions released from orthodontic devices (prosthesis, implants, wires).
  2. Figure 1 – I think it will useful to add on the figure the chemical interaction between CHDG and the metallic ions/ metallic chlorides.
  3. Line 133 – [ref]? add the reference number.
  4. Lines 105-106: The reproducibility of these diffusion coefficients are similar to those observed for other systems; i.e. within ± 2%. – you should add at least two references to sustain this affirmation.
  5. Line 226-228: Solutions for the diffusion measurements were prepared using Millipore-Q water (specific resistance = 1.82 × 105 Ω m, at 298.15 K) – the same data from lines 204-205: All solutions were freshly prepared at 15 K before each experiment – the same as line 217.
  6. Line 217: lactic acid, instated of acid lactic.
  7. Try to improve the quality of figure 3.

Author Response

Reply to the comments on the manuscript Ref.ijms-1486292  

Manuscript title: Effect of cobalt and chromium ions on the chlorhexidine digluconate as seen by intermolecular diffusion

05.December.2021

We are grateful to the Reviewer#2 for her/his time and positive appreciation on our ms. All raised comments raised have been addressed as described below (at blue for the sake of clarity).

Reviewer#2

  1. I find the introduction a little bit unclear. Actually, from line 49 to line 73, it’s not clearly highlighted the aim of the study. Try to reorganize it. Also, it is very important to mention in the introduction the novelty of the study, because there are also other studies in the literature focused on the effect of mouthwashes on metallic ions released from orthodontic devices (prosthesis, implants, wires).

We really thank the reviewer for this suggestion; we hope that the modification done in the Introduction fit the reviewer’s comment.

The Introduction in lines 49-69 reads now:

The release of metal ions from orthodontic devices in the presence of mouthwashes is a matter of concern given the potential toxicity raise and function loss of devices [9–11].

Among those devices, prosthetic restorations prosthetic restorations are subject to fretting corrosion and wear [12–14], the released wear debris may produce ionic species which have a potential toxicity depending on their concentration [15]. Consequently, it is of utmost importance to analyse the behavior of those ions in the oral cavity. There are several studies to identify released elements or determine ion concentrations [2,15]. However, at the best of our knowledge, the use of mass transport by diffusion to assess the interaction between metal ions and mouthwashes is an innovative approach.

In the present manuscript the interaction between metal ions, in particular divalent cobalt and trivalent chromium ions, and a pharmacological molecule: chlorhexidine digluconate (C34H54Cl2N10O14, Fig. 1) (CHDG), which is present in many mouthwashes is evaluated by measuring intermolecular diffusion coefficients using the Taylor dispersion method [16–18]. We intend to conclude about the potential of CHDG to act as a carrier of those ions, facilitating their removal from the oral cavity and, thus, reducing its potential toxicity.

  1. Figure 1 – I think it will useful to add on the figure the chemical interaction between CHDG and the metallic ions/ metallic chlorides.

A structure has been added to address the reviewer’s comment.

  1. Line 133 – [ref]? add the reference number.

We really apologize for this missing; a reference related with the screening effect has now been added.

  1. Lines 105-106: The reproducibility of these diffusion coefficients are similar to those observed for other systems; i.e. within ± 2%. – you should add at least two references to sustain this affirmation.

As requested two new references have been added.

  1. Line 226-228: Solutions for the diffusion measurements were prepared using Millipore-Q water (specific resistance = 1.82 × 105 Ω m, at 298.15 K) – the same data from lines 204-205: All solutions were freshly prepared at 15 K before each experiment – the same as line 217.

Thank you for pointing this out. The statement in lines 226-228 was deleted

  1. Line 217: lactic acid, instated of acid lactic.

The correction has been done.

  1. Try to improve the quality of figure 3.

The Figure 3 has been redraw and resolution improved.

Round 2

Reviewer 2 Report

The authors have taken into account all my recommendations. The article can be
published in its current form.